# A Pilot Interaction Analysis of Gut Microbiota and Peripheral Markers of Aging in Severe Psychiatric Disorders: A Role for *Lachnoclostridium*?

**DOI:** 10.3390/ijms242417618

**Published:** 2023-12-18

**Authors:** Mirko Manchia, Pasquale Paribello, Claudia Pisanu, Donatella Congiu, Athos Antoniades, Paris Vogazianos, Federica Tozzi, Federica Pinna, Aristos Aristodimou, Paola Caria, Tinuccia Dettori, Daniela Virginia Frau, Cristina Cocco, Barbara Noli, Concetta Panebianco, Valerio Pazienza, Bernardo Carpiniello, Alessio Squassina

**Affiliations:** 1Unit of Psychiatry, Department of Medical Sciences and Public Health, University of Cagliari, 09127 Cagliari, Italy; pasqualeparibello@gmail.com (P.P.); fede.pinna73@gmail.com (F.P.); bcarpini@iol.it (B.C.); 2Unit of Clinical Psychiatry, University Hospital Agency of Cagliari, 09127 Cagliari, Italy; 3Department of Pharmacology, Dalhousie University, Halifax, NS B3H 4R2, Canada; 4Unit of Neuroscience and Clinical Pharmacology, Department of Biomedical Sciences, Section of Neuroscience and Clinical Pharmacology, University of Cagliari, 09042 Monserrato, Italy; claudia.pisanu@unica.it (C.P.); dcongiu@unica.it (D.C.); squassina@unica.it (A.S.); 5Stremble Ventures Ltd., Limassol 4042, Cyprus; athos.antoniades@stremble.com (A.A.); paris.vogazianos@stremble.com (P.V.); federica.tozzi@stremble.com (F.T.); aris.aristodimou@gmail.com (A.A.); 6Unit of Biology and Genetics, Department of Biomedical Sciences, University of Cagliari, 09042 Monserrato, Italy; paola.caria@unica.it (P.C.); dettorit@unica.it (T.D.); dvfrau@unica.it (D.V.F.); 7NEF Laboratory, Department of Biomedical Sciences, University of Cagliari, 09042 Monserrato, Italy; crcocco@unica.it (C.C.); barbara.noli@unica.it (B.N.); 8Gastreonterology Unit, Fondazione IRCCS Casa Sollievo della Sofferenza Hospital, 71013 San Giovanni Rotondo, Italy; panebianco.c@gmail.com (C.P.); pazienza_valerio@yahoo.it (V.P.)

**Keywords:** interaction, biomarkers, metabolites, inflammation, aging

## Abstract

Excessive predominance of pathological species in the gut microbiota could increase the production of inflammatory mediators at the gut level and, via modification of the gut–blood barrier, at the systemic level. This pro-inflammatory state could, in turn, increase biological aging that is generally proxied by telomere shortening. In this study, we present findings from a secondary interaction analysis of gut microbiota, aging, and inflammatory marker data from a cohort of patients with different diagnoses of severe mental disorders. We analyzed 15 controls, 35 patients with schizophrenia (SCZ), and 31 patients with major depressive disorder (MDD) recruited among those attending a community mental health center (50 males and 31 females, mean and median age 46.8 and 46.3 years, respectively). We performed 16S rRNA sequencing as well as measurement of telomere length via quantitative fluorescence in situ hybridization and high-sensitivity C-reactive protein. We applied statistical modeling with logistic regression to test for interaction between gut microbiota and these markers. Our results showed statistically significant interactions between telomere length and gut microbiota pointing to the genus *Lachnostridium*, which remained significantly associated with a reduced likelihood of MDD even after adjustment for a series of covariates. Although exploratory, these findings show that specific gut microbiota signatures overexpressing *Lachnoclostridium* and interacting with biological aging could modulate the liability for MDD.

## 1. Introduction

Alterations in gut microbiota have been associated with the risk of severe mental disorders (SMD) including major depressive disorder (MDD) and schizophrenia (SCZ) [1,2]. A recent meta-analysis highlighted the presence of a transdiagnostic pro-inflammatory pattern of microbiota signatures in SMD [3]. Specifically, depleted levels of anti-inflammatory butyrate-producing bacteria *Faecalibacterium* and *Coprococcus* and enriched levels of pro-inflammatory species *Eggerthella* were consistently shared between SCZ and MDD [3]. For instance, a higher abundance of *Faecalibacterium* has been associated with decreased severity of depressive symptoms in MDD [4,5]. Similarly, *Coprococcus* was depleted in depression, as shown by a large population cohort study [6]. Further, *Eggerthella* has a key role in the metabolism of bile acids [7,8], which appear to be impaired in patients with MDD [9] and SCZ [10]. Specifically, lower levels of secondary bile acid such as taurodeoxycholic acid were associated with illness status and negatively correlated with symptom severity in both disorders [9,10]. Importantly, the presence of decreased levels of secondary bile acids has been correlated with higher levels of inflammation in in vitro models [11], and might increase the permeability of the gut [12], a factor that permits translocation of lipopolysaccharides (LPS) from the gut into the circulation [13]. This molecule can, in turn, determine a systemic low-grade inflammation via the activation of the immune system [13]. Interestingly, MDD patients with recent suicide attempts, which occurs more frequently in patients with severe psychiatric disorders such as MDD and SCZ than the general population, showed altered levels of markers of the integrity of the gut–blood barrier [14], which in turn showed also a significant correlation with pro-inflammatory interleukin 6 (IL-6). 

Inflammation is also a key component of accelerated aging which is a frequent occurrence in severe psychiatric disorders, as shown by the excess morbidity and mortality observed in these conditions [15]. Indeed, patients with severe psychiatric disorders show up to 15 years of decreased life expectancy compared to the general population [15]. Part of this excess mortality is explained by the increased risk of medical ailments in patients with severe psychiatric disorders [16], as well as by the evidence that mental disorders themselves increase the risk of a poor clinical outcome in physical diseases [17]. This bidirectional association between psychical and mental disorders has several biological mediators including genetic predisposition [18,19], increased inflammatory loading [20], and telomere shortening [20,21]. The latter is considered a specific hallmark of cellular aging. In humans, telomeric DNA is constituted by multiple (TTAGGG)n repeats ending in a single-stranded overhang of the G-rich 3′ strand and telomere-associated shelterin proteins [22,23,24]. This structure determines the integrity of the genome, ultimately protecting the chromosomes from end-to-end fusions, misrepair, and degradation [22]. Accelerated telomere shortening has been demonstrated in patients diagnosed with severe psychiatric disorders such as SCZ [25] and MDD [26]. Our group has previously shown that patients with SCZ and MDD have higher levels of inflammation and increased telomere shortening, which are both more pronounced in the more severe forms of these disorders, i.e., treatment-resistant ones [20]. Moreover, analysis of the gut microbiota composition in these cohorts showed significant differences in SCZ [27] and MDD [28] when compared to healthy controls (HCs). 

Nevertheless, little is known about the interplay between psychiatric disorders and markers of inflammation, accelerated aging, and variation in gut microbiota. It is biologically plausible that these could be the driving factors accentuating psychopathological symptoms in severe psychiatric disorders [29]. Here, we present findings from an interaction analysis of gut microbiota, aging, and inflammatory marker data from a cohort of patients with different diagnoses of severe mental disorders which were previously and separately published by our group [20,27,28].

## 2. Results

### 2.1. Characteristics of the Sample

The sample analyzed in this work was previously described elsewhere [20]. Here we present an interaction analysis based on data from individuals for whom all molecular measures were available (Table 1). Age and sex were significantly different among groups. Smoking and alcohol consumption were significantly more prevalent among SCZ and MDD patients as compared with HCs. Body mass index (BMI) was significantly higher and physical activity was lower in SCZ and MDD patients compared to HCs. No significant differences emerged in the family history of mental disorders or cardiometabolic comorbidities. Dieting styles were also not statistically different between groups (data in Appendix A). Considering the pilot nature of our investigation, in the following sections we reported findings from the models either including or not the effect of covariates. The results of the controlled models are presented in Section 2.2.3. 

### 2.2. Interaction Analysis

High-sensitivity C-reactive protein (hsCRP) was significantly different among groups (F_2_ = 2.947, *p* = 0.05) while telomere length (TL) was not significant (F_2_ = 0.267, *p* = 0.767. In the interaction analysis with microbiota, hsCRP did not yield any statistically significant results across all considered taxa and the disorders (either MDD or SCZ, case–control status). However, we found several statistically significant interactions with TL and bacterial organisms’ relative abundance across taxa in fecal samples. The methodology for interaction analyses between continuous variables, 16S metagenomics, and disease status has been recently published [30], but, to our knowledge, this is the first application of interaction testing between 16S metagenomics and blood biomarkers in psychiatric disorders. Considering the pilot nature of our investigation, the findings were not corrected for multiple comparisons. 

#### 2.2.1. Interaction Analysis in MDD

The phylum *Actinobacteria* showed a significant interaction with TL and was associated with MDD (OR = 1.005, 95% CI [1.000, 1.010]; *p* = 0.042). At the genus level, *Lachnoclostridium* showed a significant interaction with TL and was associated with HCs (OR = 0.981, 95% CI [0.968, 0.994]; *p* = 0.005). These results are illustrated in Table 2.

#### 2.2.2. Interaction Analysis in SCZ

At the family level, the interaction of *Veillonellaceae* with TL was associated with HCs (OR= 0.996, 95% CI [0.993, 0.999]; *p* = 0.02). At the genus level, *Dialister* showed a significant interaction with TL in SCZ (OR= 0.996, 95% CI [0.992, 0.999]; *p* = 0.024). All these results are outlined in Table 3.

#### 2.2.3. Adjusting for Possible Bias-Inducing Covariates

To check the internal validity of our tests, we considered possible covariates in our analysis. A Mann–Whitney U test was performed on the independent groups, HCs, MDD, and SCZ. Among the variables described in Section 2.1 showing significant differences among groups, only age was statistically significant (*p* = 0.045) between HCs (median = 37.7) and MDD (median = 54.5), whereas only BMI was significantly different (*p* < 0.001) between HCs (median = 22.7) and SCZ (median = 27.1). Finally, only BMI was significantly different (*p* = 0.002) between HCs (median = 22.7) and SCZ (median = 26.4). After introducing the significant covariates into the binary logistic regression model, only the model with MDD and HCs at the genus level based on the interaction between the relative abundance of bacterium *Lachnoclostridium* and TL remained significant (OR = 0.985, 95% CI [0.973, 0.996]; *p* = 0.001). These results are illustrated in Table 4.

## 3. Discussion

Here, we presented the findings of a secondary analysis aimed at identifying statistically significant interactions between levels of markers of accelerated aging (TL) and inflammation (hsCRP) and variation in gut microbiota in modulating the risk of SCZ and MDD. We did not find any significant interaction between levels of hsCRP and gut microbiota. However, we found statistically significant interactions between TL and gut microbiota, including the genus *Lachnostridium*, which remained significant even after adjustment for covariates. 

Our analysis showed that *Actinobacteria* significantly interacted with TL in MDD. A recent systematic review showed that most gut microbiota studies in MDD have found an increase in the relative proportion of *Actinobacteria* in patients with MDD compared to HCs (four out of five studies) [31]. Interestingly, the study by Chen et al. [32] found an increase in the relative abundance of Actinobacteria in older patients with MDD compared to age-matched HCs, presumably with accelerated aging and therefore shorter telomere length. This would be in line with our finding that TL interacted with Actinobacteria in MDD. At the genus level, we found that *Lachnoclostridium* interacted significantly with the TL in HCs compared to MDD. Age was identified as a potential covariate and therefore the interaction model was adjusted for age, remaining significant. This finding is consistent with a recent microbiome-wide analysis in 1333 individuals of the Rotterdam study showing an association (replicated in a large independent sample) of *Lachnoclostridium* with depressive symptoms [33], and with the recent multi-ethnic analysis by Bosch et al. [34]. Interestingly, and in line with the interaction identified with TL, this genus is associated with immune-inflamed gene expression profiles in tumors [35].

The phylum *Veillonellaceae* was found to significantly interact with TL in SCZ. This is in line with a number of findings [36,37] showing the association of this phylum with SCZ. *Dialister* is a genus of the family of *Veilonellaceae* indicating potentially that the observer effect at the phylum level is driven by these bacteria at the genus level. Of interest, a relatively large case–control study found that the abundance of *Dialister invisus*, which is a key driver of tyrosine biosynthesis, was positively associated with measures of cognition in patients with SCZ [38]. Although cognition was not tested in our study sample, it is generally assumed that it declines with age, a result that would be in line with our interaction findings.

These findings should be interpreted in the context of a series of limitations. First, and mainly, the cross-sectional design did not allow us to establish causality (i.e., whether the interaction between TL and microbiota is relevant for the onset of MDD, or results from the development of an illness-related pathological process). The relatively small sample size could have impacted the statistical power leading to a decreased sensitivity and specificity of our findings. Nonetheless, covariates were included, and we reported both the unadjusted and adjusted findings. It is worth noting that these types of exploratory studies are necessary to provide evidence of the potential magnitude of the main effects as well as interaction effects and enable us to design future experimental studies, including performing more accurate power analyses. In this context, this study should be considered as a hypothesis generator, exploring the validity of a multimodal approach with interaction testing in modulating the risk of MDD and SCZ. Finally, dietary habits, although accurately collected at the moment of stool specimen sampling, were not under a standardized protocol, leading to substantial heterogeneity. 

## 4. Materials and Methods

### 4.1. Sample

The molecular data analyzed in this study were available for a sample comprising 15 HCs, 35 patients with SCZ, and 31 patients with MDD recruited among those attending the community mental health center of the Unit of Psychiatry of the Department of Medical Science and Public Health, University of Cagliari and University Hospital Agency of Cagliari, and the Unit of Clinical Pharmacology, University Hospital Agency, Cagliari, Italy. The sample characteristics are reported in Table 1. The recruitment process was based on the inclusion and exclusion criteria described in Manchia et al. [39]. Briefly, the diagnosis was made according to DSM-IV criteria and the Structured Clinical Interview for DSM IV-TR Axis I Disorders (SCID) [40]. Exclusion criteria comprised the presence of acute infections, chronic autoimmune inflammatory conditions, diagnosis of any eating disorders, post-traumatic stress disorder, substance use disorders, neurological disorders, traumatic brain injury, or severe medical conditions (such as cancer or HIV infection). All patients had been followed up longitudinally with periodic assessments (typically monthly) of their psychopathological status, which included the use of standard psychometric tools in certain cases, since the illness onset. HCs had no personal or familial history of psychiatric disorders in the first degree and were recruited based on the same exclusion criteria described for patients. HCs were administered the Italian version of the SCID-I/NP 26 to rule out the presence of Axis I psychiatric disorders [40]. Patients and controls were all from the same geographical area (the island of Sardinia, Italy), Caucasians, and of Italian origin. The research protocol followed the principles of the Declaration of Helsinki and was approved by the Ethics Committee of the University of Cagliari, Italy (approval number: 348/FC/2013 and PG/2018/11693). All participants signed informed written consent after a detailed description of the study procedures.

### 4.2. Sampling Procedures

Fasting blood was collected in the morning using tubes containing Eparine for the Q-FISH experiments, or ethylenediamine tetraacetic acid (EDTA: 1.5 mg/mL) for the ELISA experiments. Blood was processed for the different protocols within 1 h of collection, during which time it was stored at a controlled temperature (4 °C). At blood drawn, patients with MDD were in the euthymic phase with an interval of at least six months from the last mood episode, meeting the diagnostic criteria. Similarly, patients with SCZ were sampled after at least six months from the last psychotic episode. Each study participant provided a fresh stool sample in a tube containing a DNA stabilization buffer (Canvax Biotech, Valladolid, Spain), from which DNA was extracted following the manufacturer’s instructions of the QIAamp DNA Stool Mini Kit (Qiagen, Germantown, MD, USA). DNA was checked for concentration and purity and stored at −30 °C until use.

### 4.3. Sequencing of Bacterial 16S rRNA Gene

An amount of 10 ng of each fecal DNA underwent library preparation for 16S rRNA gene amplicon sequencing on an Illumina MiSeq device, as previously described [41]. Briefly, the V3-V4 region was amplified, samples were barcoded, and libraries were pooled in equimolar concentrations and subjected to 2 × 300 paired-end sequencing. The de-multiplexed reads generated by MiSeq were processed using the 16S metagenomics GAIA 2.0 software (http://www.metagenomics.cloud, Sequentia Biotech 2017; Benchmark of Gaia 2.0 using published datasets available online at: http://gaia.sequentiabiotech.com/benchmark, both accessed online on 21 May 2020), quality-controlled (i.e., trimming, clipping, and adapter removal) based on FastQC and BBDuk, and mapped with BWA-MEM against the custom databases (based on NCBI) to obtain the taxonomic profile of each sample, as previously published [27]. Read pairs were processed to obtain the taxonomic profile of each sample.

### 4.4. Peripheral Markers

#### 4.4.1. Quantitative Fluorescence In Situ Hybridization (Q-FISH)

As described in Squassina et al. [20], Q-FISH was adopted to quantify the target repetitive hexameric sequences (TTAGGG) located at the distal end of chromosomes. The protocol was described in detail in Ref. [20]. 

#### 4.4.2. hsCRP

Each relevant aliquot was thawed once, immediately before running the respective ELISA assay. Plasma levels of the C-reactive protein (CRP) were measured with high-sensitivity ELISA sandwich kits (Origene: catalog number EA100881), as described in detail in Ref. [20]. The detection limit of the ELISA kits used was 0.02–3.3 mg/L.

### 4.5. Statistical Analysis

To test for interaction between the three variables (illness status, relative abundance of bacteria, and hsCRP or TL) a binomial logistic regression model was used in Ref. [42] with a level of statistical significance set at 0.05. The methodology for interaction analyses between continuous variables, 16S metagenomics, and disease status is described in detail in Ref. [30]. The Mann–Whitney U test and chi-square test were used to compare clinical, socio-demographic, and biological variables among the independent groups “HCs”, “MDD” and “SCZ” as appropriate. The statistically different variables among groups were included as covariates in the interaction tests to adjust for possible intercorrelations. 

## 5. Conclusions

This pilot analysis showed that variation in the gut abundance of *Lachnoclostridium* interacted significantly with TL in MDD. Specifically, we found that telomere length interacted with the genus *Lachnostridium* in reducing the likelihood of MDD even after adjustment for a series of covariates. These results should be considered exploratory given their secondary nature and the cross-sectional study design. At the same time, these findings point to the validity of using different sets of data for the analysis of risk estimates in severe psychiatric disorders.

## Figures and Tables

**Table 1 ijms-24-17618-t001:** Comparison of the main clinical and sociodemographic characteristics of the three samples (MDD, SCZ, and HCs).

Variable	Level	MDD (N = 31)	SCZ (N = 35)	HCs (N = 15)	*p*-Value
Age (years)	Mean (SD)	51.9 (12.9)	46.8 (12.2)	36.3 (9.1)	<0.001 *
Sex—N (%)	Males	9 (29.0)	31 (88.6)	10 (66.7)	<0.001 ^#^
	Females	22 (71.0)	4 (11.4)	5 (33.3)	
BMI (Kg/m^2^)	Mean (SD)	25.2 (5.0)	27.1 (4.2)	22.4 (2.2)	<0.001 *
Family history of mental disorders—N (%)	No	12 (41.4)	18 (51.4)	10 (66.7)	0.280 ^#^
	Yes	17 (58.6)	17 (48.6)	5 (33.3)	
Smoking habits—N (%)	Non-smoker	16 (51.6)	8 (22.9)	11 (73.3)	0.008 ^#^
	Smoker	9 (29.0)	20 (57.1)	2 (13.3)	
	Ex-smoker	6 (19.4)	7 (20.0)	2 (13.3)	
Drink habits—N (%) ^	None	17 (54.8)	23 (65.8)	3 (21.4)	0.137 ^#^
	One occasional drink	12 (38.7)	11 (31.6)	10 (71.4)	
	1–2 drinks per day	1 (3.2)	1 (2.9)	1 (7.1)	
	more than 1/2 L per day	1 (3.2)	0 (0.0)	0 (0.0)	
Physical activity—N (%)	No	20 (64.5)	23 (65.7)	4 (26.7)	0.024 ^#^
	Yes	11 (35.5)	12 (34.3)	11 (73.3)	
Cardiometabolic comorbidities—N (%)	No	23 (74.2)	23 (65.7)	13 (86.7)	0.305 ^#^
	Yes	8 (25.8)	12 (34.3)	2 (13.3)	
Age at onset (years)	Mean (SD)	35.5 (7,13)	25.6 (7.6)	NA	<0.001 *
Disease duration (years)	Mean (SD)	16.4 (12.3)	20.8 (11.9)	NA	0.144 *
History of suicide attempts—N (%)	No	25 (80.6)	27 (77.1)	NA	0.77 ^#^
	Yes	6 (19.4)	8 (22.9)	NA	

Abbreviations: FGA = first-generation antipsychotic; FISH = quantitative fluorescent in situ hybridization; HCs = healthy controls; hsCRP = high-sensitivity C-reactive protein; IQR = interquartile range; MDD = major depressive disorder; N = number; NA = not available; SCZ = schizophrenia; SGA = second-generation antipsychotic; TGA = third-generation antipsychotic. * ANOVA or Kruskal–Wallis for non-normally distributed data; # chi-square test; ^ 1 missing value for HCs.

**Table 2 ijms-24-17618-t002:** Interaction analysis of TL with bacteria according to different levels of taxa in MDD versus HCs.

Phylum	Interaction			
	*p*	OR	CI Low	CI High
*Actinobacteria*	0.042 *	1.005	1.000	1.010
**Genus**	**Interaction**			
	*p*	OR	CI Low	CI High
*Lachnoclostridium*	0.005 *	0.981	0.968	0.994

* Statistically significant *p*-value. Abbreviations: CI: confidence interval; OR: odds ratio.

**Table 3 ijms-24-17618-t003:** Interaction analysis of TL with bacteria according to different levels of taxa for the prediction of illness status in SCZ versus HCs.

Family	Interaction			
	*p*	OR	CI Low	CI High
*Veillonellaceae*	0.02 *	0.996	0.993	0.999
**Genus**	**Interaction**			
	*p*	OR	CI Low	CI High
*Dialister*	0.024 *	0.996	0.992	0.999

* Statistically significant *p*-value. Abbreviations: CI: confidence interval; OR: odds ratio.

**Table 4 ijms-24-17618-t004:** Interaction analysis of TL with the genus *Lachnoclostridium* for the prediction of illness status in MDD versus HCs including age as a covariate.

TL				Age				Bacterium				Interaction			
*p*	OR	CI Low	CI High	*p*	OR	CI Low	CI High	*p*	OR	CI Low	CI High	*p*	OR	CI Low	CI High
0.053	1.02	0.999	1.04	<0.001 *	1.017	1.01	1.03	0.001 *	3.28	1.36	7.92	0.001 *	0.985	0.973	0.996

* Statistically significant *p*-value. Abbreviations: CI: confidence interval; OR: odds ratio.

## Data Availability

The data presented in this study are available on reasonable request from the corresponding author.

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
