# Peer review of "A Pilot Interaction Analysis of Gut Microbiota and Peripheral Markers of Aging in Severe Psychiatric Disorders: A Role for Lachnoclostridium?"

_ijms, 2023, doi:10.3390/ijms242417618_

Round 1
Reviewer 1 Report
Comments and Suggestions for Authors
The authors analyze the pilot interaction of gut microbiota and peripheral markers of aging in severe psychiatric disorders by using 16S rRNA sequencing as well as measurement of telomere length via quantitative fluorescence in situ hybridization, and high sensitivity C reactive protein determination. The issues examined in this paper are interesting and relevant, however, the following issues need to be addressed before being accepted:
(1) In lines 26-27, "16S RNA" should read "16S rRNA", please revise.
(2) In Table 1 in lines 98-99, Sex-N's MDD for Female's 71.0 is missing parentheses, please add “)”, Drink habits-N's SCZ is miscalculated and HC sample size is missing one, Physical activity-N's MDD sample size is 31 or SCZ sample size is 35? please revise.
(3) Missing table underline below the Actinobacteria line in Table 2, lines 126-127, please revise.
(4) Please move the Material Methods section in line 201 to precede the Results section in line 88.
(5) Line 228, capitalize "L" in "1.5 mg/ml".
(6) Line 230, delete the space between 4 and ℃.
(7) Line 269, Conclusion, is too succinct and does not adequately describe the results of the experiment, please supplement.
(8) Lines 402-403 referencs is incorrectly formatted, pleaset revise.
Author Response
The authors analyze the pilot interaction of gut microbiota and peripheral markers of aging in severe psychiatric disorders by using 16S rRNA sequencing as well as measurement of telomere length via quantitative fluorescence in situ hybridization, and high sensitivity C reactive protein determination. The issues examined in this paper are interesting and relevant, however, the following issues need to be addressed before being accepted.
- R) We thank the reviewer for his/her positive assessment of our work.
Q1) In lines 26-27, "16S RNA" should read "16S rRNA", please revise.
R1) Revised
Q2) In Table 1 in lines 98-99, Sex-N's MDD for Female's 71.0 is missing parentheses, please add “)”, Drink habits-N's SCZ is miscalculated and HC sample size is missing one, Physical activity-N's MDD sample size is 31 or SCZ sample size is 35? please revise.
R2) We thank for highlighting these typos. We have checked and revised Table 1. There was 1 missing value for drink habits in healthy controls which has now been indicated.
3) Missing table underline below the Actinobacteria line in Table 2, lines 126-127, please revise.
R3) Revised.
4) Please move the Material Methods section in line 201 to precede the Results section in line 88.
R4) Thank you. The structure of the paper (with Results anteceding Discussion and Methods) reflects the formatting guidelines of IJMS.
5) Line 228, capitalize "L" in "1.5 mg/ml".
R5) Revised
6) Line 230, delete the space between 4 and ℃.
R6) Revised
7) Line 269, Conclusion, is too succinct and does not adequately describe the results of the experiment, please supplement.
R7) We added the following sentence: “Specifically, we found that telomere length interacted with the genus Lachnostridium in reducing the likelihood of MDD even after adjustment for a series of covariates.”
8) Lines 402-403 reference is incorrectly formatted, please revise.
R8) Revised
Reviewer 2 Report
Comments and Suggestions for Authors
The introduction (what is known) should be shorter in the abstract section. It should be max two sentences.
SCZ, MDD abbreviations should be defined in the abstract and information about the gender of patients (% of M and F) is important as same as the median/mean of age participants
In the abstract, the authors mentioned: „significant interactions between telomere length and gut microbiota pointing to the genus Lachnostridium.” – please characterize this interaction (direction, etc.) – you could replace one sentence in the introduction with more characteristics of the results.
The conclusions in the abstract are too general. Please specify more critical of your results.
Abbreviations BMI, ELISA should be defined.
In the study results, the authors should also examine the differences between groups of examined variables (the authors mentioned this issue in the Statistical Analysis section). Please add this information.
Due to differences between important variables such as BMI, age, and duration of illness, the relationship between them and biological markers should also be analyzed.
Please add information about the detection limits of using ELISA kits
In the results, the „risk” should be used carefully – please analyse https://www.ncbi.nlm.nih.gov/pmc/articles/PMC4640017/
Author Response
1) The introduction (what is known) should be shorter in the abstract section. It should be max two sentences.
R1) This has been revised.
2) SCZ, MDD abbreviations should be defined in the abstract and information about the gender of patients (% of M and F) is important as same as the median/mean of age participants
R2) This has been added to the abstract.
3) In the abstract, the authors mentioned: „significant interactions between telomere length and gut microbiota pointing to the genus Lachnostridium.” – please characterize this interaction (direction, etc.) – you could replace one sentence in the introduction with more characteristics of the results.
R3) This has been added.
4) The conclusions in the abstract are too general. Please specify more critical of your results.
R4) Done.
5) Abbreviations BMI, ELISA should be defined.
R5) Done.
6) In the study results, the authors should also examine the differences between groups of examined variables (the authors mentioned this issue in the Statistical Analysis section). Please add this information.
R6) This comparison has been added in the Results section: “High-sensitivity C-reactive protein (hsCRP) was significantly different among groups (F2=2.947, p=0.05) while telomere length (TL) was not significant (F2=0.267, p=0.767).”
7) Due to differences between important variables such as BMI, age, and duration of illness, the relationship between them and biological markers should also be analyzed.
R7) Thank you for this observation. This was performed in our analysis and detailed in section 2.2.3. Please find the modified version: “To check the internal validity of our tests we considered possible covariates in our analysis. Mann Whitney U test was performed on the independent groups HC, MDD, SCZ. Among the variables described in section 2.1 showing significant difference among groups, only age was statistically significant (p = 0.045) between HC (Median = 37.7) and MDD (Median = 54.5), whereas only BMI significantly different (p < 0.001) between HC (Median = 22.7) and SCZ (Median = 27.1). Finally, only BMI was significantly different (p = 0.002) between HC (Median = 22.7) and SCZ (Median = 26.4). After introducing the significant covariates into the binary logistic regression model, only the model with MDD and HC at the genus level based on the interaction between relative abundance of bacterium Lachnoclostridium and TL remained significant (OR=0.985, 95%CI [0.973, 0.996]; p = 0.001).”
8) Please add information about the detection limits of using ELISA kits
R8) This has been added: “The detection limit of the ELISA kits used was 0.02-3.3 mg/L.”
9) In the results, the „risk” should be used carefully – please analyse https://www.ncbi.nlm.nih.gov/pmc/articles/PMC4640017/
R9) We agree and have revised the text accordingly.